# Efficient One-Shot Neural Architecture Search With Progressive Choice Freezing Evolutionary Search

## Abstract

Neural Architecture Search (NAS) is a fast-developing research field to promote automatic machine learning. Among the recently populated NAS methods, one-shot NAS has attracted significant attention since it greatly reduces the training cost compared with the previous NAS methods. In one-shot NAS, the best candidate network architecture is searched within a supernet, which is trained only once. In practice, the searching process involves numerous inference processes for each user case, which causes high overhead in terms of latency and energy consumption. To tackle this problem, we first observe that the choices of the first few blocks that belong to different candidate networks will become similar at the early search stage. Furthermore, these choices are already close to the optimal choices obtained at the end of the search. Leveraging this interesting feature, we propose a progressive choice freezing evolutionary search (PCF-ES) method that gradually freezes block choices for all candidate networks during the searching process. This approach gives us an opportunity to reuse intermediate data produced by the frozen blocks instead of re-computing them. The experiment results show that the proposed PCF-ES provides up to 55% speedup and reduces energy consumption by 51% during the searching stage.

## 1 Introduction

Neural Architecture Search (NAS) has been proposed and extensively studied as an efficient tool for designing state-of-the-art neural networks (Elsken et al., 2019; Wistuba et al., 2019; Ren et al., 2020). NAS approaches automate the architecture design process and can achieve higher accuracy compared to human-designed architectures (Liu et al., 2019; Xie et al., 2019; Cai et al., 2019). However, the early NAS methods, such as reinforcement NAS (Zoph & Le, 2016), came with the price of expensive computation costs since every searched architecture needs to be trained from scratch, which makes the total search time unacceptable. To reduce the search cost of earlier NAS methods, the weight sharing technique has been proposed (Yu et al., 2020; Chen et al., 2020), among which the one-shot NAS method has attracted a lot of attention recently (Bender et al., 2018; Li et al., 2020).

The one-shot NAS method is known as cost-efficient as it requires training a *supernet* only once. A supernet is a stack of basic *blocks*, each of which contains multiple *choices*. A candidate network architecture (defined as *subnet*) can be formed by selecting one choice for each block in the supernet, and its corresponding weights can be inherited from the supernet. During the architecture searching stage, candidate architectures are evaluated on the validation dataset and the best architecture, i.e., the architecture with the highest validation accuracy, is updated in every searching epoch of Evolutionary Algorithm (EA) (Real et al., 2019). Surprisingly, although training is commonly deemed as a lengthy and energy-consuming task, the architecture searching stage in one-shot NAS is much more costly (Cai et al., 2020) than training a supernet. The reason is that a new searching stage should be performed whenever a different searching scenario is given, e.g., different hardware constraints, learning tasks, and workloads, while the trained supernet can be reused. Hence, the numerous inferences on the subnets can take a much longer time than training a supernet only once. According to (You et al., 2020), searching can be 10 GPU days longer than supernet training when 10 different constraints/platforms are required.

To tackle this problem, our work first makes a key observation that, for the first few continuous blocks of the candidate architecture (defined as continuous shallow blocks), their optimal *choices* can be determined at an early search epoch. Based on this observation, we propose to **freeze the choices of continuous shallow blocks at the early search epoch**, which means these choices will not be changed during the remaining search epochs. This strategy elaborately "creates" redundant computations in the continuous shallow blocks since all candidates will share exactly the same architecture, inherited weights, and input validation data for the shallow blocks during later search epochs. Then we leverage such *redundancy* and propose a simple yet effective data reuse scheme to save large amounts of computations, thus further reducing time and energy cost. Specifically, we propose to reuse the last output of the continuous shallow blocks instead of re-computing it repeatedly throughout the remaining searching stage. Interestingly, we further discover that the freezing strategy may in turn help to determine the optimal choices of the subsequent blocks earlier. Such phenomenon enables us to keep freezing the choices of blocks progressively after the initial freezing, which will create more redundant computations of the blocks that possess the same architecture (choice), thus more computations can be saved.

With the proposed freezing technique, the intermediate data (the last output of the continuous shallow blocks) of a certain subnet can be stored and reused during the evaluation of other subnets. However, as the searching stage requires to evaluate a subnet with a large batch (e.g., batch size = 5000) of input samples, storing the intermediate data of only one subnet may cause serious memory issues (Sec.3.5). Inspired by the *Importance Sampling* technique employed in many training methods (Zeng et al., 2021), we propose to sample the "important" input data that contribute more to *distinguish* the evaluation accuracy of the candidate subnets. More importantly, we empirically demonstrate that the important samples are shared across different subnets. Therefore, it only requires to store the intermediate data of important samples for one certain subnet, and then reuse it for all other subnets.

We evaluate the proposed method on multiple benchmarks trained with the state-of-the-art approaches on the ImageNet dataset (Krizhevsky et al., 2012). The experimental results indicate superb performance in improving the search efficiency while maintaining the search performance with only 0.1% searching accuracy loss. Our contributions can be summarized as follows:

- · We observe that, in the one-shot NAS evolutionary searching stage, the optimal architecture of shallow blocks is determined at the early searching stage.

- · We propose to freeze the choices of continuous shallow blocks for all candidates at the early stage, and progressively freeze the choices of the subsequent blocks in the later stage. This approach creates a great amount of redundant computations, which provide us a good opportunity to reuse the intermediate data and reduce the searching time.

- · To alleviate memory capacity issue for storing intermediate data, we leverage the concept of importance sampling and propose a distinguish-based sampling method to reduce the size of the intermediate data.

- · We conduct extensive experiments on different benchmarks with our proposed methods. The evaluation results show that our method can achieve up to 55% time saving and 51% energy saving with 0.1% accuracy loss.

## 2    REVIEW OF ONE-SHOT NAS

Different from the traditional neural network training that aims to optimize weights given a network architecture, NAS seeks to optimize both weight and architecture at the same time. Conventional NAS methods (Zoph & Le, 2016; Baker et al., 2016; Zhong et al., 2018; Zela et al., 2018) have tried to solve these two optimization problems at a nested approach. However, these methods are usually prohibitively expensive because each architecture sampled from the search space has to be trained from the scratch and evaluated separately. Recent works (Bender et al., 2018; Pham et al., 2018; Cai et al., 2019; 2018) have proposed a weight sharing strategy to reduce high costs of architecture and weight searching procedure in conventional NAS. As one of the most popular weight sharing techniques, one-shot NAS achieves unprecedented search efficiency by decoupling the whole searching process into two stages: supernet training (Fig.1 (a))and subnet searching (Fig.1 (b)). One-shot NAS encodes the search space into a *supernet* and trains it only once. Then it allows

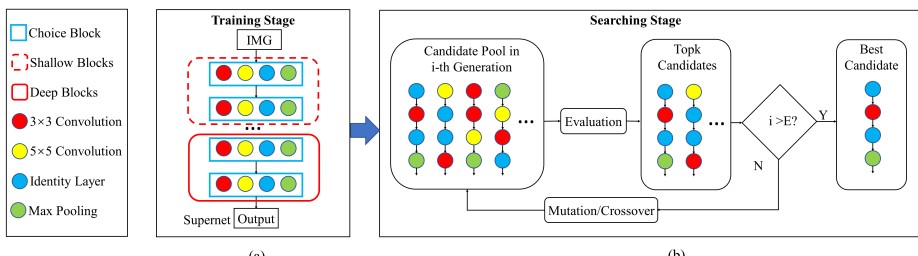

Figure 1: An Overview of One-Shot NAS.

sampled architectures directly inherit the weights from the supernet. As shown in Fig.1 (a), the commonly used chain-structured supernet (Suganuma et al., 2017) is a stack of blocks, and each block consists of multiple choices. Each choice can be different operations, e.g., a 3×3 / 5×5 convolutional layer, a max-pooling layer or an identity layer. We further define the first few blocks in the supernet as shallow blocks, while the subsequent blocks are defined as deep blocks.

The second step of one-shot NAS is the searching stage which employs EA to find the *subnet* architecture with the highest validation accuracy. Each subnet can be obtained by selecting one choice for each block in the supernet, and its weights are inherited from the supernet. The overall workflow is shown in Fig.1 (b). At each search epoch, also known as *generation*, there are $N$ candidate subnets in the candidate pool sampled from the supernet and evaluated using validation dataset. Then the candidate subnets are ranked together with the previous top candidates according to the validation accuracy. Candidates with Top-$K$ validation accuracy are selected and then evolved into $N$ new subnets by performing mutation and crossover operations (Real et al., 2019). The above process is repeated until reaching the maximum number of search epochs $E$, and the subnet with the highest accuracy can be obtained. The architecture of this subnet will be regarded as the optimal architecture since the validation accuracy obtained by using inherited weights is highly predictive on the accuracy obtained by training from scratch (Bender et al., 2018).

Recently, many studies focused on improving the efficiency and accuracy of the supernet training stage by introducing various subnet sampling methods (Guo et al., 2020; Chu et al., 2021; You et al., 2020). Nevertheless, none of them considered the efficiency of the searching stage. The searching stage can be more time-consuming compared with supernet training (Cai et al., 2020). This is because the supernet only need to be trained once while numerous searching processes are required to search the optimal architecture for various deployment scenarios, e.g., different hardware platforms, workloads. Therefore, it is essential to improve the search efficiency. NASA (Ma et al., 2021) is the first work that focuses on accelerating the search process, and a NAS accelerator was proposed that utilizes network fusion based on the computation sharing and data reuse within a search generation. Different from their approach, we propose an algorithm-level optimization for one-shot NAS evolutionary search and exploit data sharing from both within generation and across generation levels, which achieves significant improvement on the searching efficiency.

## 3 METHODOLOGY

This section first introduces several key observations during the evolutionary searching phase, based on which the progressive choice freezing evolutionary search (PCF-ES) is then proposed, along with how it saves the computations during the searching stage. Finally, to mitigate the memory issue of our method, the distinguish-based importance sampling method is proposed to reduce the size of the intermediate data.

### 3.1 OBSERVATION: MAJORITY CHOICE IN CONTINUOUS SHALLOW BLOCKS

Figure 2 illustrates how the choices of blocks evolve across generations during the evolutionary search in SPOS (Guo et al., 2020). The right-up subgraphs record the choice evolution in shallow blocks. Especially, we take the 1st, the 2nd, and the 3rd blocks as the examples to show their choice evolution across the search generations, respectively. The right-bottom subgraphs are the choice evolution in deep blocks, and the 15th, the 16th, and the 17th blocks are used as the examples. Each curve in the subgraph indicates the percentage of subnets among all candidate subnets in a certain generation that select a certain choice for the block. As an example shown in the left part of the figure, at the i-th generation, the percentage of the blue choice in the 1st block equals to 80%

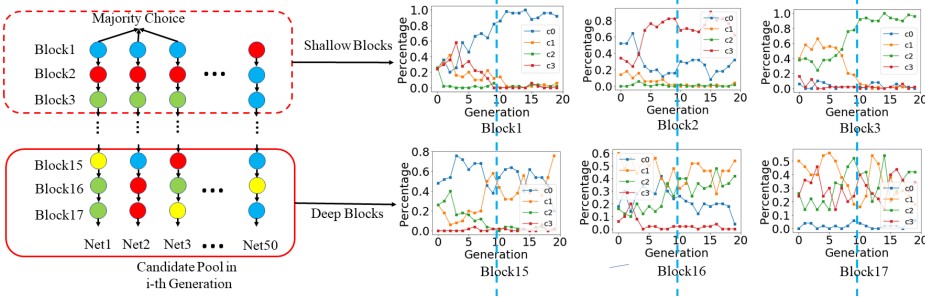

Figure 2: The percentage of different choices with generation from shallow blocks to deep blocks in SPOS.

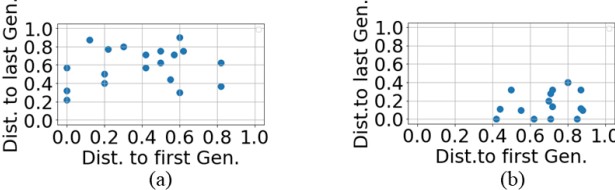

Figure 3: The distance of the shallow blocks (a) before the majority choice appears and (b) after the majority choice appears to the best architecture in the first and last generation.

if 40 out of 50 subnets select the blue choice. As can be seen in the right-up subgraphs, there is one choice whose percentage increases dramatically and surpasses all the other choices in the early generations ($i \leq 10$, highlighted by the left of the blue dotted line). We define such choice as the *majority choice*. We further observed that the majority choice will be selected by more subnets and become even more dominating for the shallow blocks in the following generations. However, this is not the case for the deep blocks. In the right-bottom subgraphs, we observed that either the majority choice emerges at very later generations, or multiple curves intertwine with each other, thus there is no obvious majority choice for deep blocks. The above observations indicate that candidate subnets tend to vote majority choice in the shallow blocks at the early searching stage, while for deeper blocks the majority choice is unclear.

## 3.2 RELATION BETWEEN THE MAJORITY CHOICE AND THE OPTIMAL CHOICE

Majority choice is the choice that most of the candidate subnets "think" it can lead to better accuracy during evolution. Intuitively, majority choice should be representative of optimal choice. To quantitatively analyze the relationship between the majority choice and the optimal choice, we first define a binary categorical choice distance $\mathcal{D}$, where $\mathcal{D}$ equals to 0 if two choices are the same otherwise $\mathcal{D}$ equals to 1. Then we calculate four types of $\mathcal{D}$ for the choices in the *continuous shallow blocks* as follows: the average distance $\overline{\mathcal{D}}$ between the non-majority choices and the block choices in the first generation (i.e., initial block choices), the average distance $\overline{\mathcal{D}}$ between the non-majority choices and the block choices in the last generation (i.e., optimal block choices), the average distance $\overline{\mathcal{D}}$ between the majority choices and the block choices in the first generation, and the average distance $\overline{\mathcal{D}}$ between the majority choices and the block choices in the last generation. Here, all $\overline{\mathcal{D}}$ denotes the averaged distance across all shallow blocks. Note that the non-majority choices are all the choices of shallow blocks when their average percentage is below an empirically predetermined threshold (0.7) at the early searching stage. The majority choices are obtained once we observe their average percentage rising above 0.7. The results of non-majority choices and majority choices are shown in Figure 3 (a) and (b), respectively. The x-axis (y-axis) represents the distance between the non-majority/majority choices and the block choices of the first (last) generation. Each point represents an independent searching process with random seeds. As can be seen, for non-majority choices (Figure 3 (a)), the points are scattered in a symmetrical way, which means a certain non-majority choice can be either close or far away from the optimal choice. In contrast, for majority choices (Figure 3 (b)), most points are located at the right-bottom of the plane, which indicates that the majority choices are very close to the final optimal choices.

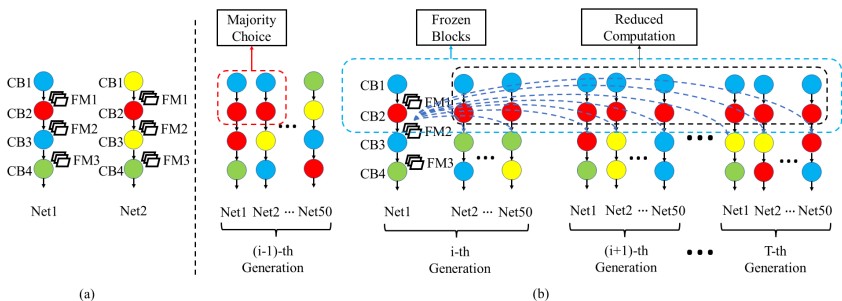

Figure 4: (a) No data reuse if the first two blocks have different choices (b) The overall workflow with choice freezing.

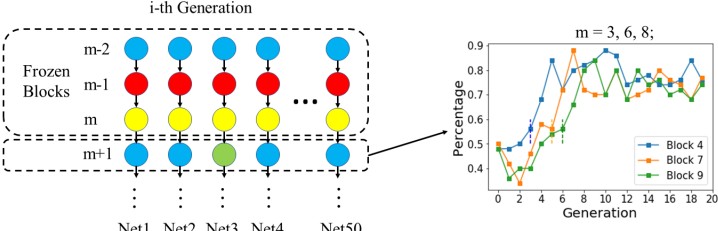

Figure 5: The choice percentage of one subsequent block after its previous blocks are frozen.

### 3.3 KEY INSIGHTS: CHOICE FREEZING

The above comparative experiment shows that majority choices are very close to optimal choices and can be safely used to approximate optimal choices. This provides a solid foundation for us to freeze the majority choices in the shallow blocks when they appear in the early searching stage. The motivation of freezing the block choices is illustrated in Fig.4. As shown in Fig 4 (a), assuming there are two candidate subnets to be evaluated sequentially at the current searching epoch, the choices of the first two blocks *CB1*,*CB2* of *Net1* and *Net2* are {*blue, red*} and {*yellow, red*} respectively. Since the choices of these two shallow blocks are not totally the same (even though the choices of *CB2* are the same), the intermediate feature maps *(FM1, FM2)* in *Net1* and *Net2* will be different. Hence, to obtain the evaluation results for both subnets, the computations of all the blocks are required. However, as shown in Figure 4 (b), if the choices in *CB1* and *CB2* are already majority choices in the previous searching epoch and we managed to freeze them at the i-th search generation, *Net1* will have the same choices for *CB1* and *CB2* as *Net2*, so does *Net3* to *Net50* in the current generation and all other subnets in the following generations. Since one-shot NAS uses the same validation dataset as input, the *FM1* and *FM2* of these subnets will be exactly the same and the computations for *CB1* and *CB2* will become redundant. Therefore, one can simply store the *FM2* and reuse it for all the subsequent evaluation of candidate subnets, with a significant computation savings of all frozen blocks. Note that the stored *FMs* can be reused both within and across generations by every subnet evaluation process, which brings more computation savings than NASA (Ma et al., 2021).

### 3.4 PROGRESSIVE CHOICE FREEZING EVOLUTIONARY SEARCH

To further explore the behaviour of the choices of deeper blocks, we keep monitoring the choice percentage of every subsequent block once the previous blocks are frozen in the candidate subnet. As shown in Figure 5, we conduct three independent search processes where the first 3, 6, and 8 continuous block choices are frozen according to the majority choice percentage threshold. For example, the orange curve represents the choice percentage of the 7th block $B_7$ if we freeze the first 6 continuous blocks at the 5th generation (marked by the dot line). It can be seen that, for all three blocks in the figure, the choice percentage increases drastically after previous blocks are fixed. Since higher choice percentage means higher possibility that a majority choice is observed, freezing the previous block choices will help freeze the latter block choices. Based on this observation, we propose to freeze more block choices progressively when the previous ones are already frozen.

The overall idea is implemented in our proposed Progressive Choice Freezing Evolutionary Search (PCF-ES) algorithm in Alg. 1. The population of the first generation is randomly generated (Line

---

**Algorithm 1** Progressive Choice Freezing Evolutionary Search

---

**Input**: supernet weight $W_A$, population size $P$, architecture constraints $C$, max generation $T$, validation dataset $D_v$, choice percentage threshold $H$, mutation rate $r$, number of blocks $N$, monitor size $f$
**Output**: architecture with best validation performance

 1: Random initialize $P_0$
 2: $c = 0$
 3: **for** $i = 1 : T$ **do**
 4:    **if** $c = 0$ **then**
 5:       **for** $j = N : 1$ **do**
 6:          average choice percentage $\overline{M_j}$ from $B_j$ to $B_1$
 7:          block ratio $Br_j$ from $B_j$ to $B_1$
 8:          **if** $\overline{M} > H$ and $Br_j > 0.5$ **then**
 9:             $a_{Top-K}^{1 \sim j} = a_{maj}^{1 \sim j}$
10:             $c = 1$
11:             break
12:          **end if**
13:       **end for**
14:    **else**
15:       **for** $x = f : 1$ **do**
16:          average choice percentage $\overline{M_f}$ from $B_{j+1}$ to $B_{j+f}$
17:          **if** $\overline{M_f} > H$ and $Br_f > 0.5$ **then**
18:             $a_{Top-K}^{1 \sim (j+x)} = a_{maj}^{1 \sim (j+x)}$
19:          **end if**
20:       **end for**
21:    **end if**
22:    $P_{mutation} = Mutation(Topk, r, C)$
23:    $P_{crossover} = Crossover(Topk, C)$
24:    $P = P_{mutation} \cup P_{crossover}$
25:    $Acc = Inference(W_A, D_v, P)$
26: **end for**
27: Return architecture with highest accuracy

---

1) and $c$ is a flag to check if there are block choices that have been frozen (Line 4). When $c$ is 0 which implies no block choice has been frozen, we calculate and examine the average choice percentage of the first $j$ blocks at each search generation, and $j$ starts from $N$ (the number of blocks in a subnet) to 1 because we aim to freeze as many block choices as possible at the beginning (Lines 5-6). Note that the average choice percentage $\overline{M_j}$ is the mean of choice percentage from block $B_1$ to $B_j$. One obvious downside of the average choice percentage is that it cannot indicate the choice percentage of each block, thus the choices of some blocks with relatively low choice percentage may be mistakenly frozen when the averaged choice percentage across the investigated shallow blocks exceeds the threshold. We employ another metric, i.e., *block ratio*, to avoid the aggressive choice frozen. *block ratio* measures the ratio of blocks whose choice percentage exceeds the threshold. For example, if the average choice percentage of the first 10 blocks exceeds the threshold, among which only 7 of these blocks exceeds the threshold, then the *block ratio* is 70%. Only when the block ratio is above the block ratio threshold, we allow the block choices to be frozen, which ensures that most of the frozen block choices reach the required choice percentage.

If both metrics, i.e., average choice percentage and block ratio, are above the thresholds (Line 8), choices of the first $j$ blocks in all Top-$K$ subnets are changed to the current majority choices (Line 9) and become frozen. Accordingly, we set flag $c = 1$ when the first $j$ blocks are frozen. For the rest of the searching process, these block choices will not be affected by the mutation and crossover operations. On the other hand, if any of the two metrics is below the threshold and the flag $c = 0$, the algorithm jumps to population generation and generates new candidates normally (Lines 22-24). If the first $j$ blocks can be frozen, we keep exploring chances to freeze more block choices during the later searching stage. The *monitor size $f$* is the maximum number of blocks we should monitor at each generation in the latter searching stage (Line 15). The purpose of using monitor size here is to prevent aggressive block freezing. Specifically, since the search space is already remarkably narrowed down in the initial freezing step (up to 8 continuous shallow blocks can be frozen at a time, while a supernet usually has a total of 20 blocks), keep freezing too many subsequent blocks (even if their average choice percentage and block ratio are above the threshold) can make the search space so small that finally leads to inaccurate search results. In the progressively freezing phase, the

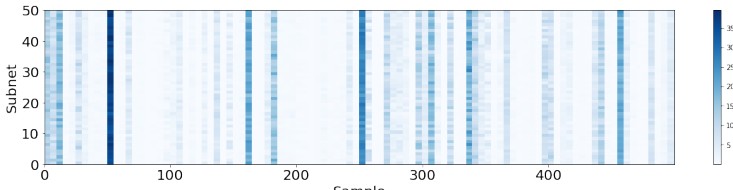

Figure 6: The importance of samples for different subnets after shallow blocks are frozen.

average choice percentage $\overline{M_f}$ is the mean of choice percentage from block $B_{j+1}$ to $B_{j+f}$ (Line 16). If the $\overline{M_f}$ is above the threshold, blocks $B_{j+1}$ to $B_{j+f}$ in Top-$K$ subnets will be frozen based on the current majority choice.

After all candidate subnets are generated, they are evaluated on the validation dataset and ranked based on the validation accuracy (Line 25). We keep track of the Top-$K$ subnets and return the Top-1 subnet architecture when maximum generation is reached or all block choices are frozen.

### 3.5 DISTINGUISH-BASED IMPORTANCE SAMPLING

To achieve the potential computation savings, it's necessary to store the intermediate feature maps, e.g. FM2 in Fig.4, of all the input samples in the validation dataset, the size of which may exceed the memory capacity of the GPUs in some circumstances. For instance, for each input sample, the intermediate data size of the second block of subnets in SPOS on ImageNet is approximately 0.8 MB. If the validation dataset contains more than 5000 input images, the memory storage of all the intermediate feature maps could exceed a GPU's memory capacity with 4GB main memory and need to be saved in CPU memory. However, transferring data from CPU memory to GPU is very time and energy consuming with compared to transferring from GPU local memory. The frequent CPU-GPU data transfer impairs the search efficiency significantly. To tackle this problem, we first leverage *Importance Sampling* (Alain et al., 2015) to compress the intermediate data. The idea of importance sampling comes from the fact that not all the input samples contribute to *distinguishing* the evaluation accuracy of candidate subnets. For example, for images which are easy to be classified by most subnets, they barely contribute to distinguishing the performance of the searched subnets.

Inspired by the importance sampling during training (Zeng et al., 2021), we follow the similar principle to sample the important data for the subnets evaluation. Let $x$ denotes the input data, $p$ denotes the uniform distribution referred from random sample, $q$ denotes the distribution adopted by importance sampling and $f(x)$ denotes the evaluation accuracy of input data $x$. The unbiased estimation of evaluation accuracy can be obtained by Eq.(1).

$$E_p[f(x)] = \int p(x)f(x)dx = E_q[\frac{p(x)}{q(x)}f(x)] \tag{1}$$

if $q(x) > 0$ whenever $p(x) > 0$.

Moreover, the estimation variance of the distribution $q$ is minimized when

$$q(x)^* = \frac{1}{Z}p(x)||f(x)||_2, \quad \text{and} \quad Z = \int p(x)||f(x)||_2 dx \tag{2}$$

In our design, we use the cross-entropy loss to approximate the evaluation accuracy and then calculate the important sampling distribution $q(x)^*$ based on Eq.(2).

Although we can obtain the important samples for a certain subnet using the above method, we cannot reuse the intermediate data of these important samples for other subnet's evaluation unless the important samples are shared across different subnets. Therefore, we quantitatively analyze the importance of samples for *different* subnets after the shallow blocks are frozen. As shown in Fig.6, we present the importance distribution of 500 important samples (x-axis) for 50 different subnets (y-axis). The color represents the importance of a sample. It can be found that the sample with high importance (deep blue) for one subnet is also important for other subnets. Overall, the importance distribution of a certain subnet (a certain row) is similar to the importance distribution of other subnets (other rows). As a result, it is only necessary to store the intermediate data of important samples for one subnet and then reuse it for other subnets.

Table 1: Comparison of searched architecture w.r.t different benchmarks and search methods. $T$: majority choice percentage threshold. $MS$: monitor size.

| Supernet | Method | T | MS | Top-1 | Top-5 | FLOPS(M) | GPU Hour | GPU Energy(MJ) |
|---|---|---|---|---|---|---|---|---|
| SPOS | Evolutionary | ✗ | ✗ | 73.5 | 90.2 | 328 | 16.7 | 4.5 |
| | CF-ES | 0.7 | ✗ | 73.5 | 90.1 | 324 | 13.9 | 3.8 |
| | PCF-ES | 0.7 | 3 | 73.4 | 90.1 | 316 | **7.5** | **2.2** |
| FairNAS | Evolutionary | ✗ | ✗ | 72.0 | 89.8 | 322 | 15.2 | 4.2 |
| | CF-ES | 0.5 | ✗ | 72.2 | 90.0 | 323 | 11.7 | 3.4 |
| | PCF-ES | 0.4 | 2 | 71.9 | 90.1 | 329 | **8.4** | **2.5** |
| GreedyNAS | Evolutionary | ✗ | ✗ | 72.5 | 90.1 | 322 | 17.9 | 4.8 |
| | CF-ES | 0.4 | ✗ | 72.3 | 90.2 | 327 | 15.1 | 3.9 |
| | PCF-ES | 0.4 | 3 | 72.5 | 90.2 | 328 | **11.1** | **3.0** |

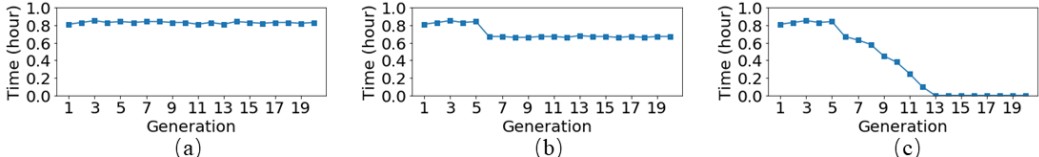

Figure 7: The search time breakdown on SPOS with (a) evolutionary search (b) CF-ES (c) PCF-ES.

## 4 EXPERIMENT

### 4.1 EXPERIMENT SETUP

Our experiments are conducted on widely used ImageNet(Krizhevsky et al., 2012), of which the validation dataset contains 50000 images at 1000 categories. Both supernet training and subnet searching processes are performed on Tesla P100 GPUs with 16 GB memory. We employ the chain structure supernet adopted with three benchmarks: SPOS (Guo et al., 2020), FairNAS (Chu et al., 2021) and GreedyNAS (You et al., 2020), whose supernets contain 20, 19 and 21 blocks, respectively. The supernets are trained with the parameter settings in the original papers. The maximum search generation is 20 and the population size is 50. At each generation, subnets with top-10 validation accuracy are kept as parent networks.

### 4.2 EXPERIMENT RESULTS

The search results on different benchmarks and search methods are shown in Table 1. Evolutionary represents the original evolutionary algorithm used in (Guo et al., 2020), which is regarded as the baseline in our study. Choice Frozen Evolutionary Search (CF-ES) represents the method that only freezes shallow blocks and does not monitor the deep blocks for further freezing. Thanks to the reduced computations for the frozen shallow blocks, CF-ES method achieves averagely 19% time saving and 18% energy saving while maintaining Top-1 and Top-5 accuracy compared to the baseline. Progressively Choice Freezing Evolutionary Search (PCF-ES) is our proposed method that freezes both shallow and deep blocks during the searching, which achieves much more latency reduction with little accuracy loss. Specifically, PCF-ES reduces 38% ∼ 55% search latency and 37% ∼ 51% GPU energy consumption in three benchmarks. To understand where the speedup exactly comes from, we further breakdown the execution time at each generation when using different searching methods. As shown in Fig.7, the searching time (latency) of each generation is close to each other in the original evolutionary search, while CF-ES takes less time since the 6th generation because shallow blocks are frozen. The latency of PCF-ES begins to decrease from the 6th generation and keeps dropping progressively until it becomes 0 in the 13th generation, during which all blocks are frozen. In other words, it takes only 13th searching epochs to find the optimal architecture when using PCF-ES method.

Fig.8 shows the Top-1 searching accuracy during the searching stage for different approaches and benchmarks. It can be observed that, for all benchmarks, all three methods have the same accuracy at the early generations when no choices are frozen. Compare to other two methods, PCF-ES shows very little accuracy loss in SPOS and FairNAS, and can always terminates earlier.

To determine appropriate hyper-parameters, i.e., choice percentage threshold, block ratio threshold, and monitor size, used in our algorithm, we conduct sensitivity analysis for all three types of thresholds. Note that the proper thresholds we obtained using one dataset are effective for different searching scenarios. We first study the effects of selecting the choice percentage threshold for dif-

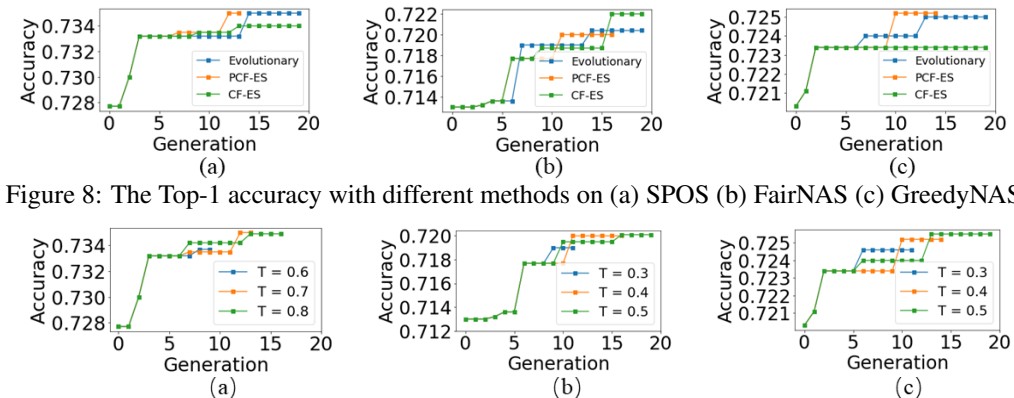

Figure 8: The Top-1 accuracy with different methods on (a) SPOS (b) FairNAS (c) GreedyNAS.

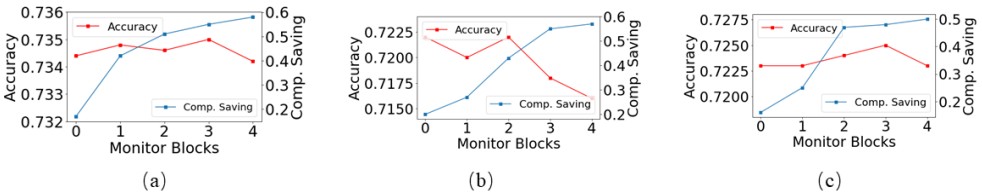

Figure 9: Top-1 accuracy v.s. generation with different majority choice percentage thresholds (T) on (a) SPOS (b) FairNAS (c) GreedyNAS.

ferent benchmarks are shown in Fig.9. The monitor size and block ratio are empirically initialized to be 3 and 0.5. As can be seen, for SPOS, when choosing the smallest threshold (T = 0.6), the searching terminates at the earliest (at the 10th generation) with relatively low accuracy. This is because some shallow blocks are mistakenly frozen. Similar behavior can be observed when setting threshold as 0.3 for FairNAS and GreedyNAS. To avoid the blocks being frozen too aggressively, a proper threshold should be picked. For example, when setting threshold as 0.7 for SPOS and 0.4 for FairNAS and GreedyNAS, the searching process maintains relatively high accuracy while terminating early as well. These thresholds are the optimal thresholds that ensure both high accuracy and low latency at the same time. The effects of selecting different block ratios are not shown here since we observed similar results to the effects of the choice percentage threshold. We empirically set block ratio as 0.5 in all three benchmarks.

Figure 10: The searching accuracy and computation savings using different monitor sizes on (a) SPOS (b) FairNAS (c) GreedyNAS.

Fig.10 shows the searching accuracy and saved computations with different monitor sizes. For SPOS and GreedyNAS, the accuracy fluctuate with the increase of the monitor size. However, for FairNAS, the accuracy drops significantly when the monitor size is greater than 2. This is because too many deep blocks are frozen in this case. The saved computation increases steadily in all benchmarks as the number of monitored blocks increases.

Fig.11 shows the validation accuracy of the optimal architecture searched by PCF-ES with different importance sampling rate (the ratio of sampled data to total data). We observe that when the sample rate is higher than 40%, the accuracy impact is negligible. However, the accuracy drops rapidly if the sample rate further decreases.

Figure 11: Validation accuracy v.s. sample rate.

## 5 CONCLUSION

In this work, we profiled the evolutionary searching process of one-shot NAS and observed the key observations regarding the choice of blocks. Motivated by the majority choices that appears at the shallow blocks at the early searching stage, we proposed a progressively choice freezing evolutionary search to narrow the search space and reduce the searching time. As our evaluation results show, our method reduces averagely 46% of searching latency and 43% of energy consumption for all benchmarks while incurs only 0.1% accuracy loss.

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

# A APPENDIX

## A.1 OBSERVATION

As introduced in Section 3.1, the majority choices of shallow blocks can be determined in the early search generations while deep blocks cannot. Here we present the choice evolution of more blocks in two other one-shot NAS benchmarks: FairNAS and GreedyNAS. As shown in Fig. 12. Moreover, choice evolution of different dataset are shown in Fig.13. It can be found that the majority choices of shallow blocks appears in the early search stage and their percentage increases steadily. On the contrary, the majority choice in deep blocks can not be decided during the search process.

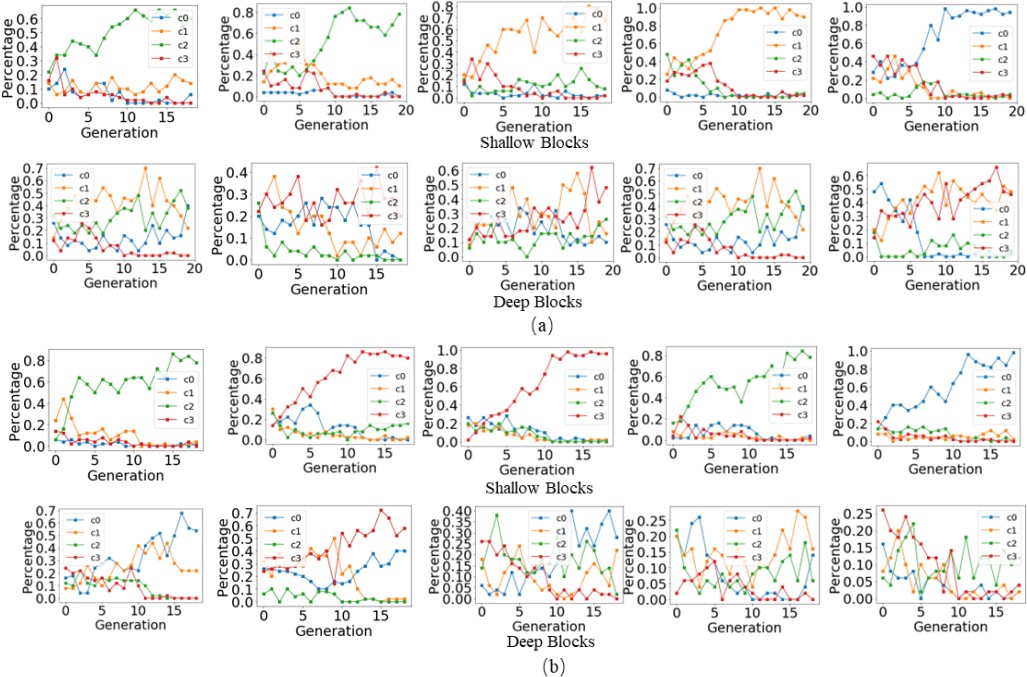

Figure 12: The percentage of different choices in shallow blocks and deep blocks v.s. generations on (a) FairNAS (b) GreedyNAS.

## A.2 PROGRESSIVELY CHOICE FREEZING

In section 3.4, we show that in SPOS, the choice percentage of the subsequent blocks increases rapidly after the previous blocks are frozen. We further conduct the same experiments for other two benchmarks. As shown in Fig. 14, for all six searching processes in (a) and (b), the choice percentage of subsequent blocks increases dramatically after freezing the previous blocks.

## A.3 EXPERIMENT RESULTS WITH MORE DATASET

We conducted our experiment on multiple datasets to support Section 4.1. Some of the experiment results on CIFAR-10 and CIFAR-100 dataset at SPOS are shown as Table 2. It is can be found that, our CF-ES method could achieve up to 20% time saving with no accuracy loss while PCF-ES achieves up to 52% time saving with negligible accuracy loss.

## A.4 THE COMPARISON BETWEEN SAMPLING METHOD

We compare the validation accuracy of the importance sampling with random sampling, as shown in Table 3. It can be found that our importance sampling method achieves higher or the same accuracy compared to random sampling at all sample rates. Our method can achieve a lower sample rate while maintaining high search accuracy.

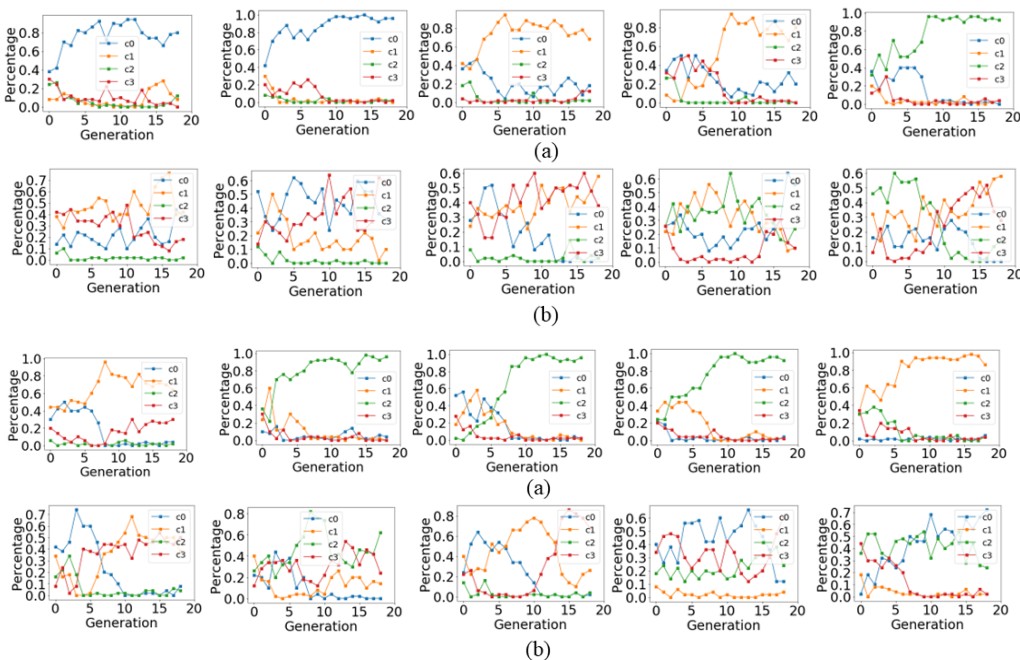

Figure 13: The percentage of different choices in shallow blocks and deep blocks v.s. generations on SPOS on (a) CIFAR-10 (b) CIFAR-100.

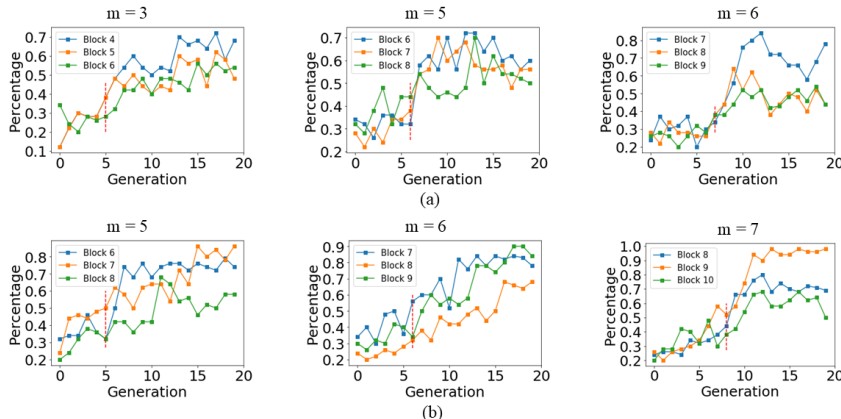

Figure 14: The choice percentage of subsequent blocks after previous blocks are frozen in (a) Fair-NAS (b) GreedyNAS.

Table 2: Comparison of searched architecture w.r.t different dataset

| Dataset | Method | Top1 | Top5 | GPU Hour |
|---------|--------|------|------|----------|
| CIFAR-10 | Evolutionary | 96.5 | 99.9 | 5.4 |
| | CF-ES | 96.5 | 99.9 | 4.6 |
| | PCF-ES | 96.4 | 99.9 | 3.0 |
| CIFAR-100 | Evolutionary | 75.1 | 92.4 | 5.4 |
| | CF-ES | 75.0 | 92.1 | 4.4 |
| | PCF-ES | 74.9 | 92.3 | 2.6 |

Table 3: Validation accuracy with random sampling and distinguish-based importance sampling

| Sample Rate / Method | 90% | 70% | 50% | 30% |
|---|---|---|---|---|
| Random Sampling | 73.4 | 73.5 | 73.1 | 73.1 |
| Importance Sampling | 73.5 | 73.5 | 73.4 | 73.1 |

Table 4: Validation accuracy with distinguish-based importance sampling and quantization.

| Bit / Sample Rate | 32 | 16 | 8 | 4 |
|---|---|---|---|---|
| 50% | 73.4 | 73.4 | 73.1 | 73.0 |
| 30% | 73.1 | 73.1 | 72.8 | 72.8 |
| 10% | 73.1 | 73.1 | 73.0 | 72.8 |

## A.5    THE EFFECTS OF QUANTIZATION

In an attempt to achieve more memory savings, we tested another compression technique, i.e., *quantization*, on our sampled dataset. Quantization is a commonly used compression method that converts a floating-point value to a fixed-point value with fewer bits. Here the quantization is applied on the intermediate data before storing them into the memory. The validation accuracy with different importance sampling rate and quantization bit for SPOS benchmark is shown in Table 4. It can be seen that the search accuracy maintains when the intermediate data are quantized to 16 bit, but drops considerably when further quantized to lower bit. For the configuration of PCF-ES method, we use 40% sample rate and 16-bit quantization for the best searching performance.

