# OpenReview forum: "Efficient One-Shot Neural Architecture Search With Progressive Choice Freezing Evolutionary Search"
_ICLR.cc/2023/Conference — Submitted to ICLR 2023_

### Official Review · Reviewer_zkYQ · 2022-10-21

**Confidence:** 5
**Correctness:** 2
**Technical Novelty And Significance:** 3
**Empirical Novelty And Significance:** 3
**Recommendation:** 3

**Clarity, Quality, Novelty And Reproducibility:**

Overall, the paper is well-written and it is an important problem. The presented idea is relatively simple but there has been little prior work in that direction making this quite original work.

There is a general lack of comparisons against reasonable baselines. First of all, a comparison against NASA is missing that was correctly identified as related work. However, other simpler ideas are not explored either:
- Cache computations on disk
- Reduce the number of generations in EA
- Reduce the validation dataset size
- Reduce the population size

Admittedly, each of these methods may have their own problems (additional hyperparameters, additional disk space, drop in predictive performance).

The hyperparameter sensitivity is a critical issue in my opinion. We neither have a theoretical nor empirical evidence that we would be able to choose the right hyperparameters for the presented method. Given that the method is only twice as fast as the hyperparameter-free baseline, retrying with a different set of hyperparameters makes this method pointless. I would like to see a strategy how to set the hyperparameters efficiently otherwise there is simply no point in using the method in the first place.

To me it is not clear why we need the importance sampling step. There is no need to keep all data in GPU memory. However, reducing the validation data itself will obviously reduce the overall time. This can be combined with the baseline as well. It is unclear, how much this contributes to the solution.

**Strength And Weaknesses:**

Strengths: important, under-explored aspect of NAS, clear description of the work
Weaknesses: lack of baselines (see next section for actionable feedback), hyperparameter sensitivity (discuss seriousness, find ways to overcome it)

**Summary Of The Paper:**

The authors propose a method which accelerates the architecture choice step of certain one-shot NAS methods. The method greedily freezes the first block choices during the architecture optimization phase with an evolutionary algorithm. This allows for caching the intermediate feature maps and therefore reduce the computational effort. In an experimental study on ImageNet, they show that this method saves time at no reduction in predictive performance. Furthermore, they investigate the hyperparameter sensitivity.

**Summary Of The Review:**

In this well-written paper, the authors present an original idea to reduce the time for finding the architecture after a one-shot NAS. The proposed method reduces time by a factor of 2 but additionally it adds several sensitive hyperparameters. The authors study this sensitivity but provide no advice how to select stable hyperparameters. This raises the question whether we are able to reduce the time or we spend additional time searching for these hyperparameters. Finally, the only existing baseline is not considered for comparison as well as other simple baselines that could accomplish the same.

---

> ### Author Response · Authors · 2022-11-18
> **Response to reviewer zkYQ**
>
> Thank you for your careful and valuable comments. We will explain your concerned weakness point by point.
>
> Q1: There is a general lack of comparisons against reasonable baselines. First of all, a comparison against NASA is missing that was correctly identified as related work. However, other simpler ideas are not explored either:
> Cache computations on disk;
> Reduce the number of generations in EA;
> Reduce the validation dataset size;
> Reduce the population size;
>
> A: NASA proposed a hardware accelerator based on network fusion within a generation. On the contrary, our method improves the efficiency from a higher level which explores data reuse from both inter-generation and intra-generation. Our method is orthogonal to their method, and our method can employ their accelerator at the hardware level if needed. So the comparison between these two methods is unnecessary.
>
> These methods such as cache computation, reduce the generation, reduce the number of generations and population size is orthogonal to our work. Our method can be integrated with these methods. Reducing the validation data directly might hurt the search accuracy.
>
> Q2: The hyperparameter sensitivity is a critical issue in my opinion. We neither have a theoretical nor empirical evidence that we would be able to choose the right hyperparameters for the presented method. Given that the method is only twice as fast as the hyperparameter-free baseline, retrying with a different set of hyperparameters makes this method pointless. I would like to see a strategy how to set the hyperparameters efficiently otherwise there is simply no point in using the method in the first place.
>
> A: The experiment results of different hyperparameters are shown in Figure 9, 10 and 11. For each supernet, the hyperparameter tuning needs to be performed only once. For a frequently employed model searched for numerous scenarios, the time of hyperparameter tuning is negligible compared to the overall searching time.
>
> Q3: To me it is not clear why we need the importance sampling step. There is no need to keep all data in GPU memory. However, reducing the validation data itself will obviously reduce the overall time. This can be combined with the baseline as well. It is unclear, how much this contributes to the solution.
>
> A: Transferring data from CPU memory to GPU is very time-consuming and energy-consuming compared to transferring from GPU local memory. The frequent CPU-GPU data transfer impairs search efficiency significantly. Reducing the validation data directly hurt the search accuracy. Our importance sampling method with PCF-ES achieves both high efficiency and negligible accuracy loss.

---

### Official Review · Reviewer_cLeE · 2022-10-24

**Confidence:** 4
**Correctness:** 4
**Technical Novelty And Significance:** 3
**Empirical Novelty And Significance:** 2
**Recommendation:** 8

**Clarity, Quality, Novelty And Reproducibility:**

The paper is clearly presented and well written. For this reviewer it was easy to follow the though process in the work.

The proposed idea is quite novel, but not earth-shattering.

It would be possible to reproduce something similar to the work performed here, though some guesswork for parts would be needed.

**Strength And Weaknesses:**

Strengths:
- The paper idea is intuitive and well worked out
- The description of the idea is well presented making the idea clear to the reader

Weaknesses:
- The paper has no clear related work section and it is hence unclear how the authors feel their work relates to prior work save for the discussion of work they base themselves on.
- The conclusion is very short and provides little in the way of insights
- The only results are on ImageNet - it would be interesting to see how the approach works on datasets not over-evaluated with NAS approaches.

**Summary Of The Paper:**

The paper introduces the idea of freezing layers within a One-Shot evolutionary NAS approach. As the lower layers quickly converge to a 'final' state the authors argue that freezing them and removing the calculations performed within them will reduce computation and hence energy. An approach is developed to identify when layers can be frozen and another approach to identify the 'significant' samples as pre-computing the computations (and re-using) can overwhelm the memory available within a GPU.

**Summary Of The Review:**

The paper is clear and concise. The idea seems reasonably novel and is supported by good evidence.

Some more specific comments:

- The review of one-shot NAS is interesting, but probably too long for this paper. More on related work would be better to present.

- There is a bizarre blue line under the middle bottom graph in figure 2.

- "This provide a solid" -> "This provides a solid"

---

> ### Author Response · Authors · 2022-11-18
> **Response to reviewer cLeE**
>
> Thank you for your careful and valuable comments. We will explain your concerned weakness point by point.
>
> Q1: The paper has no clear related work section and it is hence unclear how the authors feel their work relates to prior work save for the discussion of work they base themselves on.
>
> A: Although there is no stand-alone related work section in our paper, related works on which our work is based are reviewed and discussed in Section 2.
>
> Q2: The conclusion is very short and provides little in the way of insights
>
> A: We improved our conclusion in Section 5 and added more details.
>
> Q3: The only results are on ImageNet - it would be interesting to see how the approach works on datasets not over-evaluated with NAS approaches.
>
> A: We conducted our experiment on multiple datasets. Some of the experiment results on CIFAR-10 and CIFAR-100 dataset at SPOS are shown as follows:
> |     Dataset      |     Method          |     Top1    |     Top5     |     GPU   Hour    |
> |------------------|---------------------|-------------|--------------|-------------------|
> |     CIFAR-10     |     Evolutionary    |     96.5    |     99.9    |     5.4           |
> |                  |     CF-ES           |     96.5    |     99.9     |     4.6           |
> |                  |     PCF-ES          |     96.4    |     99.9     |     3.0           |
> |     CIFAR-100    |     Evolutionary    |     75.1    |     92.4     |     5.4           |
> |                  |     CF-ES           |     75.0    |     92.1     |     4.4           |
> |                  |     PCF-ES          |     74.9    |     92.3     |     2.6           |
> These experiment results are added in the Appendix A.3. And observation of majority choice on more datasets are added in Appendix A.1.
>
>
> Q4: Some more specific comments.
>
> A: Section 2 is relatively long because we reviewed one-shot NAS and discussed the related work of our method in the same section.
>
> Blue lines are employed to highlight the comparison between block choice percentages at the same generation.
>
> The mentioned sentence is revised as advised.

---

### Official Review · Reviewer_Gnhw · 2022-10-25

**Confidence:** 4
**Correctness:** 2
**Technical Novelty And Significance:** 2
**Empirical Novelty And Significance:** 1
**Recommendation:** 3

**Clarity, Quality, Novelty And Reproducibility:**

The algorithm is clearly described and well-motivated. The idea of progressively freezing choices may be a reasonable design choice. However, the detail of the experiments in Section 3.1 (observation) and Section 4 are not provided. Because of the lack of statistical evaluation, reproducibility is low.

**Strength And Weaknesses:**

# Strength

Energy saving is becoming a crucial recently. The proposed approach aims at contributing to this point.

# Weaknesses

Single Dataset: Evaluation is only done on a single dataset. Its generality is not discussed.

Observation: The observation in Section 3.1 seems to be observed on a single dataset as well. The experimental settings are not provided for this observation.

Comparison: The proposed approach is compared only with SPOS variants. It has not been compared with other one-shot NAS approaches such as DSNAS, which does the weight training and architecture search at once. Because of the lack of the evaluation it is not clear whether the proposed approach is useful among other one-show NAS approaches.

Statistical Evaluation: No statistical evaluation has been performed. It is not even clear what are the numbers in Table 1 and Figures. Are they average values?



**Summary Of The Paper:**

This paper focused on accelerating one-shot neural architecture search (NAS). In particular, the authors aims to accelerate the architecture search phase based on evolutionary approach. The authors first discuss their observation that the evolutionary approaches tend to select the same shallow blocks from the early stage of the search. Then, an approach is proposed that freezes the selected blocks in shallow part of the super-network. This approach has an advantage that the features computed in all population in the evolutionary approach are the same because they share the same blocks, leading to saving some computational time. Moreover, to further speedup the search process, a mechanism to subsample validation dataset to approximate the validation accuracy with low computational time. The proposed approach, PCF-ES, have be compared with a baseline evolutionary approach and a variant of the proposed approach on three supernet variants on ImageNet. Without compromising the performance significantly, saving 50% of GPU hours and GPU energy are reported.

**Summary Of The Review:**

As mentioned in the strength and weaknesses section, numerical evaluation in this paper is not sufficient to support the claim of this paper.

---

> ### Author Response · Authors · 2022-11-18
> **Response to reviewer Gnhw**
>
> Thank you for your careful and valuable comments. We will explain your concerned weakness point by point.
>
> Q1: Single Dataset.
>
> A: We conducted our experiment on multiple datasets. Some of the experiment results on CIFAR-10 and CIFAR-100 dataset at SPOS are shown as follows:
> |     Dataset      |     Method          |     Top1    |     Top5     |     GPU   Hour    |
> |------------------|---------------------|-------------|--------------|-------------------|
> |     CIFAR-10     |     Evolutionary    |     96.5    |     99.9    |     5.4           |
> |                  |     CF-ES           |     96.5    |     99.9     |     4.6           |
> |                  |     PCF-ES          |     96.4    |     99.9     |     3.0           |
> |     CIFAR-100    |     Evolutionary    |     75.1    |     92.4     |     5.4           |
> |                  |     CF-ES           |     75.0    |     92.1     |     4.4           |
> |                  |     PCF-ES          |     74.9    |     92.3     |     2.6           |
> These experiment results are added in the Appendix A.3.
>
> Q2: Observation.
>
> A: More observation data with different dataset of Section 3.1 is shown in Appendix A.1. The experiment setup is illustrated in section 4.1.
>
> Q3: Comparison.
>
> A: The DSNAS is a conventional one-stage framework that conducts the training and searching at once. However, our method focuses on improving the efficiency of the one-shot NAS searching process. The comparison between one-stage NAS and one-shot NAS is out of the scope of our method. We have employed our method at multiple one-shot benchmarks and shows the efficiency among all approaches evidently.
>
> Q4: Statistical Evaluation.
>
> A: The numbers in Table 1 are averaged values from 3 experiment. The experiment setup is discussed in Section 4.1.

---

> > ### Comment · Reviewer_Gnhw · 2022-11-24
> > **Thanks for the additional results, but they are not convincing.**
> >
> > Thank you for the response. I still have some concerns.
> >
> > What if the GPU hour is fixed for all methods? Currently reported results only show that these three methods are at different places on the Pareto front of accuracy and GPU hour.
> >
> > Because the motivation is to reduce the energy saving, one should take into account the time for the weight training as well. If one consider the overall GPU hour, is the reported difference in time between the methods meaningful? If the weight training takes significant time, why don't you just take more efficient NAS approaches than SPOS from the beginning? From this perspective, one should compare the proposed approach to other approaches applying weight sharing.

---

> > > ### Author Response · Authors · 2022-12-01
> > > **Response to reviewer Gnhw**
> > >
> > > Thank you very much for your further comments and advice. We will answer your questions point by point:
> > >
> > > Q1: What if the GPU hour is fixed for all methods? Currently reported results only show that these three methods are at different places on the Pareto front of accuracy and GPU hour.
> > >
> > > A: In our experiment, we compared the time and energy consumption at the same search generations and our method achieves significant time and energy saving with no or negligible accuracy loss. If the GPU hour is fixed at some circumstances, our method achieves much more search generation than the conventional method and a lot more search space will be explored, and a lot more candidate subnets will be evaluated.
> > >
> > > Q2: Because the motivation is to reduce the energy saving, one should take into account the time for the weight training as well. If one consider the overall GPU hour, is the reported difference in time between the methods meaningful? If the weight training takes significant time, why don't you just take more efficient NAS approaches than SPOS from the beginning? From this perspective, one should compare the proposed approach to other approaches applying weight sharing.
> > >
> > > A: A supernet architecture only needs to be trained once and can be reused in numerous later works, while the searching stage needs to be performed whenever a new constraint is given. In practice, the searching cost is much higher than supernet training in a frequently employed model. Achieving significant time and energy saving in the searching stage is important.
> > >
> > > Although there are other weight sharing methods, we chose these benchmarks because one-shot NAS is a widely accepted and widely employed approach.

---

### Official Review · Reviewer_Y3fv · 2022-11-02

**Confidence:** 3
**Correctness:** 3
**Technical Novelty And Significance:** 3
**Empirical Novelty And Significance:** 2
**Recommendation:** 3

**Clarity, Quality, Novelty And Reproducibility:**

* The motivation and approach of the proposed method are reasonable.
* The proposed method might be novel and is based on interesting observations. However, it is unclear whether it works well on other datasets and tasks because the experiment is conducted using only the ImageNet dataset.
* The experimental evaluation is not enough to validate the effectiveness of the proposed PCF-ES.
* Because the authors did not provide the code and the detailed experimental settings, it is hard to reproduce the experimental results.

**Strength And Weaknesses:**

**Strengths**
* An interesting observation in one-shot NAS, the first few continuous blocks of candidate architectures become similar in the early search epoch, is presented.
* The proposed PCF-ES can reduce the architecture search cost without significant performance deterioration on the ImageNet dataset.

**Weaknesses**
* The authors mention that the architecture searching cost is higher than the supernet training cost. It may be true when searching multiple architectures with different complexities by optimizing different objective functions. However, the proposed method only targets searching for a well-performed architecture. The reviewer suspects that the search cost reported in the experiment does not include supernet training. The search cost of both supernet training and architecture search phases, i.e., total search cost, should be reported in the experiment.
* The authors treat the existing work of NASA (Ma et al. 2021) as related work on accelerating the architecture search process. However, an empirical comparison of NASA and the proposed method is not performed.
* The experiment is conducted using only the ImageNet dataset. It is not clear whether the assumption that the first few blocks become similar among the candidate solutions will hold in other datasets and tasks. To evaluate the proposed method in various situations, it might be a possible choice to use NAS-Bench datasets for further performance evaluations.
* The authors consider that the intermediate feature maps to be reused should be stored on the GPU memory. The reviewer thinks that it is possible to store the feature maps on the CPU memory, although the data transferring cost occurs. It is not clear the motivation for storing all intermediate feature maps on the GPU memory.
* The detailed algorithm of importance sampling is unclear. How to calculate the probability distribution of $q(x)^*$ and the integral over $x$ in (2)? Also, the probability distribution of $p(x)$ is needed to use the equation (1). How to calculate $p(x)$ which is the distribution of input data x?
* The effectiveness of importance sampling seems to be unclear. It would be better to compare the proposed sampling method to random sampling.
* In the proposed PCF-ES, two techniques, freezing the shallow blocks and importance sampling-based input sample selection, are introduced. It is not clear which techniques mainly contribute to computational cost reduction. The ablation study should be conducted. What are the computational costs of the method only using the freezing technique and the method only using the importance sampling-based sample selection? If we store the intermediate feature maps on the CPU memory, we can evaluate the method only using the freezing technique.

**Summary Of The Paper:**

This paper proposes an efficient one-shot neural architecture search (NAS) method that progressively freezes the architectures. The authors show that the first few blocks become similar among the candidate solutions during the evolutionary architecture search and then develop an acceleration method of architecture search by exploiting this phenomenon. In addition, the input sampling method is introduced to reduce memory consumption when reusing intermediate feature maps among candidate architectures. The experimental results demonstrate that the proposed progressive choice freezing evolutionary search (PCF-ES) can speed up the architecture search compared to existing one-shot NAS methods.

**Summary Of The Review:**

The authors treat an interesting topic in one-shot NAS. However, as described in the section on Strength And Weaknesses, the reviewer feels that the weaknesses of this paper outweigh the strength.

---

> ### Author Response · Authors · 2022-11-18
> **Response to reviewer Y3fv**
>
> Thank you for your careful and valuable comments. We will explain your concerned weakness point by point.
>
> Q1: The search cost of both supernet training and architecture search phases, i.e., total search cost, should be reported in the experiment.
>
> A: The training time of all three benchmarks are shown as follows:
> |     Benchmark                     |     SPOS    |     FairNAS    |     GreedyNAS    |
> |-----------------------------------|-------------|----------------|------------------|
> |     Training   Cost (GPU days)    |     12      |     12         |     7            |
> The supernet training is time-consuming at first sight. However, a supernet architecture only needs to be trained once and can be reused in numerous later works while the searching stage need to be performed whenever a new constraint is given. In practice, the searching cost is much higher than supernet training in a frequently employed model.
>
> Q2: However, an empirical comparison of NASA and the proposed method is not performed.
>
> A: NASA proposed a hardware accelerator based on network fusion within a generation. On the contrary, our method improves the efficiency from a higher level which explores data reuse from both inter-generation and intra-generation. Our method is orthogonal to their method, and our method can employ their accelerator at the hardware level if needed. So the comparison between these two methods is unnecessary.
>
> Q3: The experiment is conducted using only the ImageNet dataset.
>
> A: We conducted our experiment on multiple datasets. Some of the experiment results on CIFAR-10 and CIFAR-100 dataset at SPOS are shown as follows:
> |     Dataset      |     Method          |     Top1    |     Top5     |     GPU   Hour    |
> |------------------|---------------------|-------------|--------------|-------------------|
> |     CIFAR-10     |     Evolutionary    |     96.5    |     99.9    |     5.4           |
> |                  |     CF-ES           |     96.5    |     99.9     |     4.6           |
> |                  |     PCF-ES          |     96.4    |     99.9     |     3.0           |
> |     CIFAR-100    |     Evolutionary    |     75.1    |     92.4     |     5.4           |
> |                  |     CF-ES           |     75.0    |     92.1     |     4.4           |
> |                  |     PCF-ES          |     74.9    |     92.3     |     2.6           |
> These experiment results are added in the Appendix A.3. And observation of majority choice on more datasets are added in Appendix A.1.
>
> Q4: It is not clear the motivation for storing all intermediate feature maps on the GPU memory.
>
> A: Transferring data from CPU memory to GPU is very time-consuming and energy-consuming compared to transferring from GPU local memory. The frequent CPU-GPU data transfer impairs search efficiency significantly.
>
> Q5: The detailed algorithm of importance sampling is unclear.
>
> A: In equation (1), $p(x)$ denotes the uniform distribution referred from random sampling and $f(x)$ denotes the cross-entropy loss of input $x$. In equation (2), $q(x)$* is derived from $Z$ which is calculated from the $p(x)*||f(x)||_2$ over all validation data.
>
> Q6: It would be better to compare the proposed sampling method to random sampling.
>
> A: The experiment comparison with importance sampling and random sampling is shown as follows:
> |     Sample   Rate    |     0.9     |     0.7     |     0.5     |     0.3     |
> |----------------------|-------------|-------------|-------------|-------------|
> |     Random   Acc     |     73.4    |     73.5    |     73.1    |     73.1    |
> |     IS   Acc         |     73.5    |     73.5    |     73.4    |     73.1    |
> It can be found that our importance sampling method achieves higher or the same accuracy with compared to random sampling at all sample rates. Our method can achieve a lower sample rate while maintaining high search accuracy. These experiment results are added in the Appendix A.4.
>
> Q7: It is not clear which techniques mainly contribute to computational cost reduction.
>
> A: We evaluate these two methods separately, the experiment results are modified with your advice. The experiment results of employing CF-ES, PCF-ES and PCF-ES with IS are shown in Table 1.
>
> | Supernet  | Method       | T   | MS | Top-1 | Top-5 | FLOPS(M) | GPU Hour | GPU Energy(MJ) |
> |-----------|--------------|-----|----|-------|-------|----------|----------|----------------|
> | SPOS      | PCF-ES w/ IS | 0.7 | 3  | 73.4  | 90.1  | 327      | 4.1      | 1.3            |
> | FairNAS   | PCE-ES w/ IS | 0.4 | 2  | 71.7  | 89.5  | 325      | 5.5      | 1.6            |
> | GreedyNAS | PCE-ES w/ IS | 0.4 | 3  | 72.2  | 90.0  | 325      | 7.2      | 1.9            |
>
> And the results of applying both freezing technique and importance sampling with different sample rate are shown in Figure 11. More experiment results are shown in Appendix.

---

### Decision · Program_Chairs · 2023-01-20

**Decision:**

Reject

**Justification For Why Not Higher Score:**

The reviewers criticized many issues in the paper, including a lack of reasonable baselines (2x faster and only a bit worse; but is this better than a reasonable baseline of half the epochs, half the data, etc, which would also yield 2x speedups?), evaluation on too few datasets without it being clear whether there are confounding factors, lack of discussion of related work, no code availability and thus unclear reproducibility.

**Justification For Why Not Lower Score:**

N/A

**Metareview: Summary, Strengths And Weaknesses:**

This paper introduces a method for speeding up the „search phase“ of one-shot NAS methods, which takes a trained supernet as input and only needs to find the best architecture for a given set of constraints. The reviewers agree that this is an understudied problem. However, they also criticized many issues in the paper, including a lack of reasonable baselines (2x faster and only a bit worse; but is this better than a reasonable baseline of half the epochs, half the data, etc, which would also yield 2x speedups?), evaluation on too few datasets without it being clear whether there are confounding factors, lack of discussion of related work, no code availability and thus unclear reproducibility. As a result of this, three reviewers gave clear rejections scores. One reviewer gave a score of 8, but their review is not tremendously positive, either, and they did not react to my request for a clarification about their score. I am therefore following the majority of the reviewers and recommend rejection. I encourage the reviewers to rethink their choice of baselines and experimental protocol. One resource that could help with this is https://jmlr.org/papers/v21/20-056.html for some best practices in NAS experimentation.